# Effects of Lemongrass Essential Oil on Key Aromas of Pickled Radish During Storage Using HS–GC–IMS and in Silico Approaches

**DOI:** 10.3390/foods14050727

**Published:** 2025-02-21

**Authors:** Zelin Li, Ziqi Gao, Chao Li, Yanghuan Wu, Yiqiu Xia, Linyu Ni, Jing Yan, Yifan Hu, Dongyu Wang, Zhirui Niu, Changwei Cao, Hao Tian, Xiuwei Liu

**Affiliations:** 1Agro-Products Processing Research Institute, Yunnan Academy of Agricultural Sciences, Kunming 650223, China; lzl1054994094@126.com (Z.L.); 13129747859@163.com (Z.G.); 18685744425@163.com (C.L.); 15187620391@163.com (Y.W.); 13320527459@163.com (Y.X.); 13668783010@163.com (L.N.); 90jing1126@sina.com (J.Y.); huyisarah@163.com (Y.H.); 2Department of Food Science and Engineering, College of Biological and Food Engineering, Southwest Forestry University, Kunming 650224, China; ccwylf1111@163.com; 3Yun’an Industrial College, Yunnan Agricultural Vocational and Technical College, Kunming 650212, China; 18849688676@163.com; 4Yunnan Institute of Product Quality Supervision and Inspection, National Tropical Agricultural By-Products Quality Inspection and Testing Center, Kunming 650223, China; bullnzr@163.com

**Keywords:** pickled radish, lemongrass essential oil, storage, key aromas, molecular docking

## Abstract

To investigate the effects of lemongrass essential oil on the key volatile aroma compounds of pickled radish (PR) during storage, this study used headspace–gas chromatography–ion mobility spectrometry, fingerprint analysis, multivariate statistical analysis, and molecular docking to study different PR samples. The results indicated that a total of 48 volatile aromatic compounds were identified. Fingerprint analysis revealed that the aroma profiles of samples at different storage stages were different. Using the screening criteria of *p* < 0.05 and variable importance for the projection > 1 in multivariate statistical analysis, and relative odor activity value > 1, six potential key aroma compounds were selected. Furthermore, phenylethyl acetate, β-ocimene, 4-heptanone, and limonene were determined as the key aroma compounds that affect the PR aroma profile after adding lemongrass essential oil. Moreover, the addition of lemongrass essential oil increased the fruit and sweet aroma of PR samples during storage. The results of molecular docking indicated that the recognition of these four odors was mainly accomplished through hydrophobic interactions and hydrogen bond interactions by docking OR1A1 and OR5M3 receptors. This study can offer a preliminary foundation and theoretical support for the in-depth exploration of the paocai industry.

## 1. Introduction

Paocai, a quintessential Chinese fermented vegetable product, boasts a rich consumption history and can be crafted from a diverse array of vegetables including Chinese cabbage, radishes, peppers, cucumbers, and ginger [1]. Pickled radish (PR) is one of the most popular paocai in traditional Chinese cuisine. During the pickling process, it undergoes a series of intricate biochemical reactions, generating unique flavors, and metabolites, causing changes in nutrient content and sensory quality, and therefore is highly favored by the Chinese people [2]. The enhancement of free fatty acids, γ-aminobutyrate, niacin, glutamate, and acetate in PR has significantly elevated its nutritional profile [3]. Some studies have revealed a diverse spectrum of unique phenolic compounds within the context of PR, endowing it with a range of bioactive functions, including antibacterial activity, antioxidant potential, modulation of gut microbiota composition, and other significant biological effects [4,5]. The content of isothiocyanates and other aromas increases, contributing to the distinctive flavor of PR [2]. For example, appropriate amounts of acetic acid and lactic acid can enhance the pleasant aroma and taste of PR [6]. Furthermore, the perception of PR aroma/taste can be attained through the molecular docking simulation of small molecules interacting with olfactory/gustatory receptors [7]. However, studies have indicated that long-term or improper storage may result in an increase in the acidity of PR, the emergence of unpleasant aromas, and even the formation of white film or other detrimental substances, ultimately causing deterioration in its quality [8,9]. This deterioration in the quality of PR can constitute an irreparable loss for industries, small workshops, or families. Although there are some countermeasures involving microorganisms that can mitigate the deterioration of PR quality during storage [8,10], additional methods are required to alter the current situation.

Lemongrass essential oil serves as a green alternative in the fields of nutrition, pharmaceuticals, and the food industry because of its antibacterial, antiviral, nematicidal, antifungal, insecticidal, and antioxidant properties [11]. Its various volatile components act against harmful microorganisms. The citral of lemongrass essential oil shows remarkable antibacterial efficacy against a wide range of bacteria by affecting the cell membranes, compromising their integrity and causing cell death [12]. It has also been reported that α-terpinene, α-pinene, myrcene, geraniol, linalool, nerol, and γ-terpinene possess very strong antibacterial properties [13]. The volatile components of lemongrass essential oil not only possess the aforementioned biological attributes but also endow it with its aromatic features. It has been extensively utilized in flavors, fragrances, and aromatherapy [11,14]. At present, there are no reports on the effects of lemongrass essential oil on key aromas of pickled radish during storage.

Therefore, this study is aims to compare and screen the effects of lemongrass essential oil on the key aroma and flavor mechanism of PR during storage using HS–GC–IMS, fingerprint analysis, multivariate statistical analysis, relative odor activity value (ROAV), and molecular docking (MD) methods. This study can offer a preliminary foundation for the application of lemongrass essential oil in the paocai industry and furnish theoretical support for the in-depth exploration of the aroma perception mechanism of PR.

## 2. Materials and Methods

### 2.1. Materials

A total of 6 kg of pickled radish samples with same size were provided by Honghe Hongbin Food Co., Ltd., Guiyang, China on 12 June 2024. These samples were randomly divided into 6 groups, each weighing 1 kg. Each group’s 1 kg sample was randomly and equally apportioned into three replicate tests. Among them, three group samples were treated with 0.1% (*w*/*w*) of lemongrass essential oil for a 28-day storage experiment. During the 0–28 day storage period, the samples were packaged in food-grade vacuum bags and stored at room temperature. Samples, both with and without added lemongrass essential oil, were collected every 14 days and subsequently rapidly frozen in liquid nitrogen for further analysis. And the samples containing lemongrass essential oil were, respectively, labeled as EOPR0, EOPR14, and EOPR28, while the other samples were, respectively, marked as PR0, PR14, and PR28.

### 2.2. Chemicals

The n-ketone standard substances (2-butanone, 2-pentanone, 2-hexanone, 2-heptanone, 2-octanone, and 2-nonone) were purchased from Sigma-Aldrich Co., Ltd. (St. Louis, MO, USA). Nitrogen (purity ≥ 99.999%) and 20 mL of headspace bottle were obtained from Shandong Hanon Scientific Instruments Co., Ltd (Shandong, China). MXT-WAX capillary column (15 m × 0.53 mm, 1.0 μm) was purchased from Restek Corporation (Bellefonte, PA, USA).

### 2.3. Headspace–Gas Chromatography–Ion Mobility Spectrometry (HS–GC–IMS) Analysis

#### 2.3.1. HS Conditions

A total of 2.0 g of samples were placed in a 20 mL headspace vial and added 10 μL of 2-methyl-3-heptanone at 10 μg/mL concentration and incubated at 50 °C for 15 min.

#### 2.3.2. GC–IMS Detection

The GC–IMS instrument (FlavourSpec^®^, G.A.S, Dortmund, Germany) equipped with a MXT-WAX capillary column (15 m × 0.53 mm, 1.0 μm; Restek Corporation, Bellefonte, PA, USA) was used to analyze the differences in aromas among pickled radishes by referring to previous study [15]. The GC conditions were as follows: the injection volume was 500 µL. Nitrogen (with a purity of no less than 99.999%) was employed as the carrier gas. The carrier gas was kept at 2.0 mL/min for 2 min, and then increased linearly to 10.0 mL/min within 8 min, and further rose to 100.0 mL/min within 10 min and maintained for 20 min. The drift gas was maintained at 150 mL/min throughout the entire process. The IMS conditions were as follows: the tritium source (^3^H) was utilized as the ionization source; the length and temperature of the migration tube were 53 mm and 45 °C, respectively. The intensity of the electric field was set at 500 V/cm in positive mode.

#### 2.3.3. Qualitative and Quantitative of Aroma Volatile Compounds

For the qualification of aroma volatile compounds, the retention index (RI) was computed through using the retention time (RT) of n-ketones (C4–C9) as the standard, and the calculated RI were compared with those of the GC–IMS library standards and the NIST 2020 database [16]. The RI was calculated as presented in Equation (1).(1)$$\mathrm{RI}=100\times (\frac{T_{v}-T_{n}}{T_{n+1}+T_{n}}+n)$$

T_v_ represents chromatographic peak RT of aromas; T_n_, T_n+1_ represent RT of n-ketones C_n_ and C_n+1_.

The relative content of each aroma volatile compound was computed in accordance with the peak area normalization method (Equation (2)).(2)$$\mathrm{Relative content}=\frac{P_{a}}{p_{o}}\times 100\%,$$

P_a_, aromas peak area. P_o_, overall peak area.

### 2.4. ROAV Calculated

The aroma volatile compound that contributes the most prominently to the overall flavor profile of the sample was determined to have a ROAV of 100, and the ROAV of other components (i) was calculated in accordance with Equation (3) [7].(3)$$\mathrm{ROAV}\approx \frac{C_{i}\times T_{\max}}{C_{\max}\times T_{i}}\times 100,$$

C_i_ represents the relative content (%) of aromas; T_i_ represents the aroma threshold in water (μg/kg); C_max_ and T_max_ stand for the relative content and threshold of the aroma which contributed the most to the sample.

### 2.5. MD Analysis

The MD analysis between the 4 key aroma volatile compounds (limonene, 4-heptanone, phenylethyl acetate M, β-ocimene M) and six olfactory receptors (OR8D1, OR7D4, OR5M3, OR2W1, OR1G1, and OR1A1) was conducted in accordance with the method depicted by Pu et al. [17]. Herein, the 3D conformer of olfactory receptors and aroma volatile compounds was downloaded from the RCSB PDB database (https://www.rcsb.org, accessed on 20 August 2024) and PubChem database (http://pubchem.ncbi.nlm.nih.gov, accessed on 20 August 2024), respectively. The software of PyMOL 2.6.0 (DeLano Scientific LLC, Palo Alto, CA, USA), AutoDockTool 1.5.7 (Scripps Research, La Jolla, CA, USA), and Discovery Studio 2019 (BIOVIA Inc., Paris, France) were utilized for conducting the MD and visual analysis of the three-dimensional (3D) and two-dimensional (2D) binding processes.

### 2.6. Statistical Analysis

All experiments were performed with three independent biological replicates. Data analysis and compilation were facilitated by the VOCal 0.4.03 software associated with GC–IMS (Flavour Spec^®^, GAS, Dortmund, Germany). Subsequent statistical processing was executed using SPSS 29.0 (IBM Corp., New York, NY, USA), where one-way ANOVA coupled with Duncan’s post hoc test was utilized for assessing differences among multiple groups. A *p*-value of less than 0.05 was considered to denote statistical significance. Additionally, principal component analysis (PCA) and orthogonal partial least squares discriminant analysis (OPLS-DA) were conducted employing SIMCA 16 (Umetrics, Umea, Sweden).

## 3. Results and Discussion

### 3.1. Analysis of Qualitative and Quantitative of Aroma Volatile Compounds

To explore the alterations of key aroma volatile compounds in diverse samples during storage, GC–IMS was employed to detect volatile compounds at various stages. The data were acquired from the six group samples, which enabled the identification of aroma volatile compounds and the discrimination of their differences during storage. Two-dimensional spectra are presented in Figure 1a. The vertical axis represents the retention time, the horizontal axis indicates the ion migration time, and the reactive ion peak (RIP) is marked by a red line located on the left side of the graph. The existence of each bright spot on the right of the RIP indicates an aroma volatile compound. A white color indicates a low content, while a red color indicates a high content. Furthermore, the intensity of color is directly proportional to the concentration of aromas [18]. As the storage time progresses, the aroma differences among different PR samples mainly exist in the RT range of 300–1200 s and the drift time range of 8–15 ms, and the intensity of aromas increases. The peak intensities of aroma compounds in the PR sample containing lemongrass essential oil were generally higher than those in the one without. To investigate differences in samples, the spectrograms of 0-day PR samples were taken as the reference, and a sample difference plot was acquired by subtracting the reference from the spectra of the other five groups (Figure 1b). Consequently, the subtracted background appeared white when aroma volatile compound concentrations were equal, blue when they fell below the reference value, and red when they exceeded it [19]. It was found that there was an increase in the concentration of aromas in sample groups EOPR0, EOPR14, and EOPR28 compared to PR samples, as indicated by the red color. Simultaneously, the concentration of a limited number of aroma volatile compounds decreased (Figure 1b).

Three-dimensional spectroscopy can enhance the differentiation of aroma volatile compound characteristics in the sample more effectively [18]. Consequently, this study conducted a comprehensive analysis of the three-dimensional spectral data from samples at various stages and corroborated the findings observed in the two-dimensional spectral data (Figure 1c). The variations in aroma observed during PR samples storage may be attributed to the inherent volatility of lemongrass essential oil, along with the volatile compounds that facilitate the metabolism of its flavor precursors [12,20].

The identification and quantification of target substances were achieved by integrating the detected odor compound signals with the GC retention index database (NIST 2020) and the IMS migration time database provided by VOCal software, alongside the utilization of internal standards for relative comparison. A total of 48 aroma volatile compounds were identified, including 9 ketones, 4 acids, 10 esters, 7 alcohols, 11 aldehydes, 6 olefins, and 1 ether. Notably, GC–IMS detected monomers or dimers forms of most of the aroma volatile compounds in the sample (Appendix A and Table 1). The concentrations of aroma compounds in these samples varied throughout the storage period (Table 1).

Ketones are stable in nature and have a pleasant, sweet smell such as fruit aroma and milk aroma [18]. During the storage period, nine ketones were found in the PR and EOPR samples. 6-Methylhept-5-en-2-one, 4-heptanone, and 1-octen-3-one showed a significant rise in relative content after adding lemongrass essential oil and increased over time. This occurrence could potentially be ascribed to the existence of lemongrass essential oil, which expedites the metabolic process of radishes (including unsaturated fatty acid oxidation/degradation, amino acid degradation) and potentiates the oxidant activity of beneficial microorganisms [12,21]. Other ketone compounds exhibited varying degrees of change during storage.

Volatile organic acids represent a category of compounds that play a crucial role in imparting the characteristic acidic flavor to paocai. Elevated concentrations of acetic acid can result in an overpowering and undesirable sour taste [22]. The acetate concentration in EOPR samples at the same storage stage was significantly lower (*p* < 0.05) compared to that in PR samples, and the rate of acetate accumulation in EOPR samples was slower than that observed in PR samples as storage time increased. This finding suggests that lemongrass essential oil inhibits the production of acetic acid in PR. Butyric acid can provide paocai with a fresh aroma [23]. In this study, 3 butyric acids were detected, suggesting that lemongrass essential oil could promote the content of butyric acid in PR samples during storage.

Esters are significant compounds found in fermented foods, typically characterized by their pleasant aromas. The fruity and floral notes of paocai originate from esters, which are predominantly produced through esterification reactions involving short-chain acids and aldehydes [24]. This study detected a total of 10 ester fragrances, and the relative content of lynalyl acetate and 2-(methylthio) acetate significantly increased after adding lemongrass essential oil. Ethyl acetate was the highest ester content throughout the storage process, although the EOPR samples showed a significant decrease compared to the PR sample in the 0–14 days, it was higher than the PR sample at 28 days of storage. This indicated that lemongrass essential oil maintained the fruity and sweet aroma provided by ethyl acetate during PR storage. In addition, alcohols and esters typically contribute to a desirable fresh and fruity flavor profile, enhancing the palatability of pickles [25]. The incorporation of lemongrass essential oil could lead to a notable increase in the concentration of most alcohols present in the PR sample during the storage period (Table 1). This phenomenon is primarily attributed to the activities of beneficial microorganisms, including lactic acid bacteria and acetic acid bacteria [26].

Aldehydes prevalently emerge during the metabolic processes of amino acids, decarboxylation of keto acids, and oxidation of unsaturated fatty acids; they constitute essential characteristic flavor constituents in paocai [24]. Alkenes can also augment the overall flavor profile of paocai [24]. Citronellal, limonene, and myrcene are reported the main components of lemongrass essential oil [12]; therefore, its content significantly increases on day 0, and then decreases as storage time increases (Table 1). These findings suggested that the incorporation of lemongrass essential oil could modify the aromatic profile of the PR sample.

### 3.2. Fingerprint Spectra Analysis of Aromas

Lemongrass essential oil has also been used in the food industry for extending shelf life and maintaining sensory quality during storage [27]. To investigate the dynamic changes in the characteristic aroma volatile compounds of different samples during storage, the identified volatile compounds were plotted into a fingerprint. The columns signify the variations in the quantity of a specific aroma volatile compound among diverse samples, and the rows represent the amount of each aroma volatile compound within the same sample. The color of a fingerprint indicates the concentration of an aroma component, and the more intense the color, the greater the quantity. As shown in Figure 2, there were three major differences between the PR and EOPR samples during storage. During storage, the main aroma volatile compounds in the PR sample were hexanal (M, D), diallyl sulfide, (*E*)-2-hexenal, and heptanal, which exhibited an increase (Figure 2 green box). After the addition of lemongrass essential oil, the concentrations of the four compounds decreased throughout the storage period (Figure 2 green box). The concentration of 21 aroma compounds in the PR sample increase rapidly, with 17 of them showing an increase as the storage time prolonged (Figure 2 purple box and red box). These results suggested that lemongrass essential oil could enhance the overall aroma profile of PR and promote the intensity of aroma volatile compounds during storage, but it inhibited the increase in the concentration of a small number of aroma volatile compounds. The difference might be attributed to the selective inhibition of microorganisms that produce hexanal (M, D), diallyl sulfide, (*E*)-2-hexenal, and heptanal from lemongrass essential oil or its effect on the metabolism of small molecule compounds in the PR sample [11,12].

### 3.3. Multivariate Statistical Analysis of Aromas

To further screen the key aroma volatile compounds, the compounds detected above underwent multivariate statistical analysis, and the results are presented in Figure 3. In the unsupervised analysis of PCA, the PR and EOPR samples are each grouped into their own clusters, suggesting that there were differences in the overall flavor profile of these samples (Figure 3a). The loading plot of PCA indicated that the compounds butyrolactone, 2,3-butanedione, 2,3-butanediol, and acetic acid might be accountable for this difference (Figure 3b). The visualization of the supervised OPLS-DA analysis further suggested that PR and EOPR samples showed distinctions during storage (Figure 3c), and the OPLS-DA model has not suffered from overfitting in 200 permutation tests (Figure 3d), thereby affirming the good explanatory and predictive ability of the classification result [28]. The key aroma volatile compounds were screened with *p* < 0.05 and variable importance for the projection (VIP) value > 1 as the criteria. The study indicated that *p* < 0.05 and VIP > 1 are significant indicators for the selection of key volatile compounds, as these volatile compounds play a potential and critical role in the overall aroma profile of the sample under these criteria [29]. As shown in Figure 3e, 13 volatile aromatic compounds were screened out, including ethyl propanoate, ethanol M, 2-methylpropanol, 2,3-butanedione, citronellal, *E*-2-octenal, benzaldehyde, acetic acid, ethyl crotonate, nonanal D, ethyl heptanoate, ethanol, and phenylethyl acetate M. Nevertheless, this was merely screened at a statistical level and their contribution to aroma requires further confirmation in combination with ROAVs.

### 3.4. Key Aromas Screening

To screen the key aroma volatile compounds of PR samples and the effect of lemongrass essential oil on them during storage, a combination of ROAV and VIP > 1 was conducted. ROAV/OAV > 1.0 is regarded as one of a selection criterion that makes a significant contribution to the overall flavor profile of paocai [30]. Table 2 listed the 12 aroma compounds that possess an ROAV > 1 during at least one stage of storage. Furthermore, the Venn diagram revealed that there were six compounds that fulfill both VIP and ROAV (at least in one storage stage) greater than 1, including 2,3-butanedione, citronellal, benzaldehyde, acetic acid, nonanal D, and phenylethyl acetate M (Figure 4a). Thus, they were regarded as potential key aroma volatile compounds for PR samples. Among them, after the addition of lemongrass essential oil, with the exception of phenylethyl acetate, the ROAV of the other five odors demonstrated a downward trend during storage (Table 2). It is reported that phenylethyl acetate M possesses a sweet, rose-like, and honey aroma, presenting apple-like fruit notes along with a delicate hint of cocoa and whiskey-like undertones [31]. The addition of lemongrass essential oil could enhance the sweet and fruity aromas in the EOPR sample during storage. Paocai deteriorates during storage as a consequence of the activities of microorganisms and metabolic reactions, which lead to the generation of excessive compounds like nitrite and organic acids, giving rise to excessive acidity or bitterness and thereby undermining its quality [32]. In this study, lemongrass essential oil was capable of reducing the ROAV of acetic acid at different stages, suggesting that it could retard the deterioration of EOPR quality. Nonanal, despite having the aromas of citrus and rose with low olfactory threshold and a high ROAV, also possesses a potent oily odor [33]. The ROAV of nonanal (M,D) in the EOPR sample declined during storage, suggesting that it could suppress the generation of oily odors attributed to nonanal. In addition, lemongrass essential oil also notably enhanced the ROAVs of limonene, 4-heptanone, and β-ocimene in EOPR, particularly with ROAV surpassing 1 after 14 days of storage. These results also indicated that lemongrass essential oil could increase the storage process of EOPR with fruity and sweet aromas.

### 3.5. Analysis of MD Results

This section employed molecular docking to investigate the molecular mechanism through which lemongrass essential oil enhanced the fruity and sweet aroma during the EOPR storage. Phenylethyl acetate, limonene, 4-heptanone, and β-ocimene were, respectively, docked with the most prevalently reported olfactory receptors (OR8D1, OR7D4, OR5M3, OR2W1, OR1G1, and OR1A1) of flower and fruity aromas [7]. As shown in Figure 4b, all these key aroma compounds could bind to olfactory receptors with lower binding energies, suggesting that the augmentation of these aroma volatile compounds was the consequence of their stimulation of multiple receptors rather than the effect of a single receptor. The lower the binding energy, the stronger the affinity between the small molecule ligand and the olfactory receptor, and the more pronounced the olfactory perception [35]. Phenylethyl acetate, β-ocimene, 4-heptanone, and limonene exhibited the lowest binding energies of −7.2, −6, −5.1, and −6.9 kcal/mol with OR1A1, OR5M3, OR1A1, and OR1A1, respectively. The fruity and sweet aroma of EOPR enhanced by the lemongrass essential oil might be dominated by OR1A1 and OR5M3, with other olfactory receptors playing a supporting role.

Visualize the 2D and 3D docking result of the complexes with the highest affinity to investigate its binding interaction mechanism. As shown in Figure 5, the four key aroma volatile compounds bind to OR1A1 and OR5M3 mainly through hydrogen bonding, van der Waals forces, pi-Alkyl, Alkyl, and pi-Sigma interactions. Phenylethyl acetate primarily bonds to OR1A1 through hydrogen bonds (bond to amino acid residues: TRY276, TRY258, TRY178) and van der Waals interactions (Figure 5a). Hydrogen bond and van der Waals force interactions are the main forms of ligand–receptor binding in small molecules [7]. Interestingly, β-ocimene, 4-heptanone, and limonene bond to many amino acid residues of OR1A1 and OR5M3 mainly through the interaction of pi-Alkyl and Alkyl. The interaction between Pi-Alkyl and Alkyl represents a form of molecular interaction via Van der Waals force, where the Pi-Alkyl interaction involves the interplay between the electron density-rich π bond and the alkyl chain [36]. Moreover, Pi-Alkyl and Alkyl belong to hydrophobic interactions. Consequently, these results suggest that lemongrass essential oil influences the perception of key aroma compounds during the storage of EOPR samples, mainly through hydrophobic interactions with the container and to a relatively lesser extent through hydrogen bond interactions.

## 4. Conclusions

This study identified 48 volatile aromatic compounds in different pickled radish samples using HS–GC–IMS, including 9 ketones, 4 acids, 10 esters, 7 alcohols, 11 aldehydes, 6 olefins, and 1 ether. Further, a total of six compounds were selected by multivariate statistical analysis, fingerprint, and ROAV to be considered as key aroma volatile compounds. Additionally, adding lemongrass essential oil could increase the ROAVs of β-ocimene, 4-heptanone, and limonene in the EOPR sample, which could enhance the sweet and fruity aroma of the PR sample during storage. The molecular docking results indicated that the four key odorant compounds selected mainly achieve odor perception through hydrophobic interactions with amino acid residues of OR1A1 and OR5M3, as well as hydrogen bonds. This study can offer a preliminary foundation for the application of lemongrass essential oil in the paocai industry and furnish theoretical support for the in-depth exploration of the aroma perception mechanism of PR.

## Figures and Tables

**Figure 1 foods-14-00727-f001:**
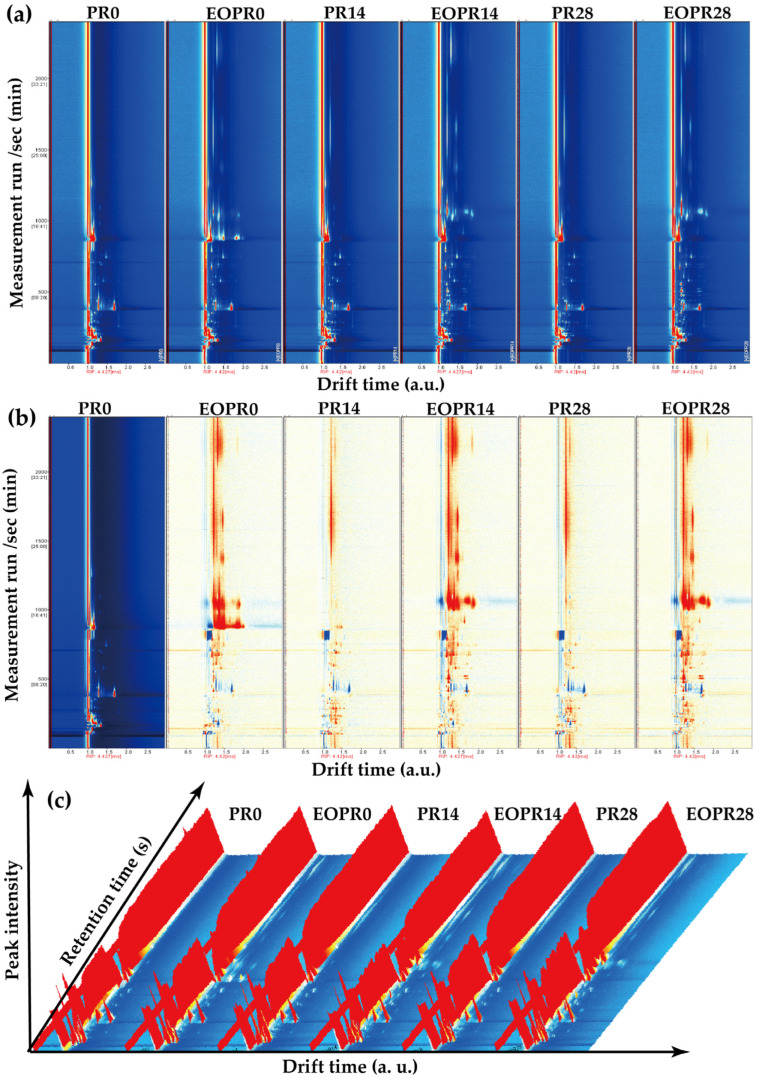
Analysis of aroma volatile compounds in different pickled radish samples. (**a**) Two-dimensional spectrum of GC–IMS; (**b**) topographic subtraction map; (**c**) three-dimensional topographic map. EOPR0, EOPR14, EOPR24, respectively, represent the pickled radish samples of adding lemongrass essential oil for 0, 14, 28 days of storage. PR0, PR14, PR24, respectively, represent the pickled radish samples for 0, 14, 28 days of storage.

**Figure 2 foods-14-00727-f002:**
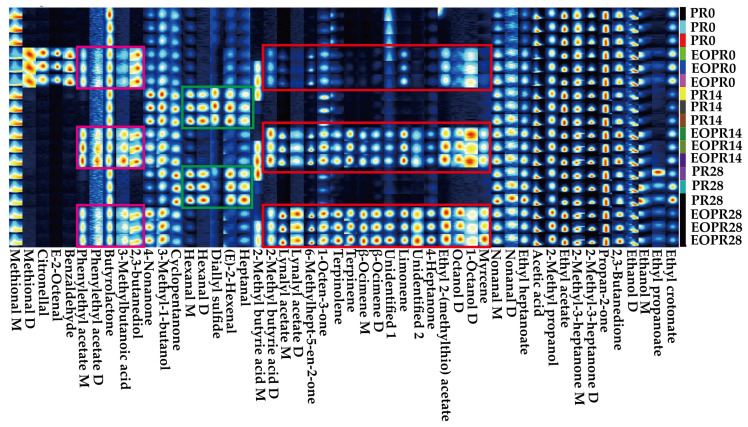
The results of fingerprint spectra analysis of with and without lemongrass essential oil pickled radish samples during the 0–28 day storage period. EOPR0, EOPR14, EOPR24, respectively, represent the pickled radish samples of adding lemongrass essential oil for 0, 14, 28 days of storage. PR0, PR14, PR24, respectively, represent the pickled radish samples for 0, 14, 28 days of storage.

**Figure 3 foods-14-00727-f003:**
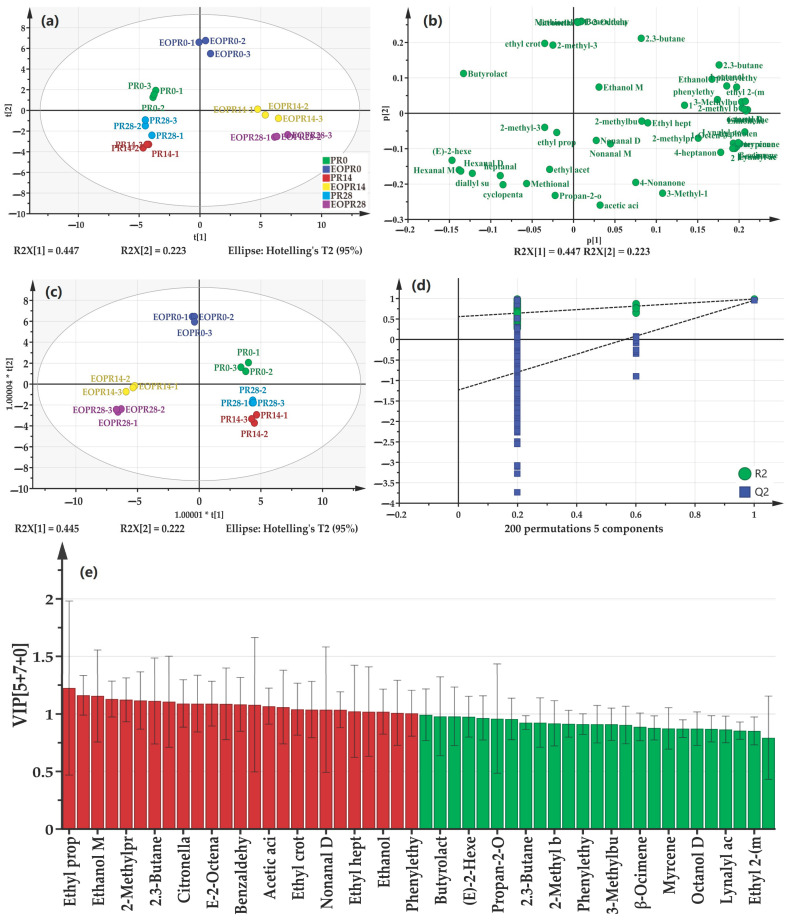
The results of multivariate statistical analysis. (**a**) PCA score plot; (**b**) loading plot; (**c**) OPLS-DA score plot; (**d**) permutation test of OPLS-DA model; (**e**) variable importance for the projection value (VIP) values. EOPR0, EOPR14, EOPR24, respectively, represent the pickled radish samples of adding lemongrass essential oil for 0, 14, 28 days of storage. PR0, PR14, PR24, respectively, represent the pickled radish samples for 0, 14, 28 days of storage.

**Figure 4 foods-14-00727-f004:**
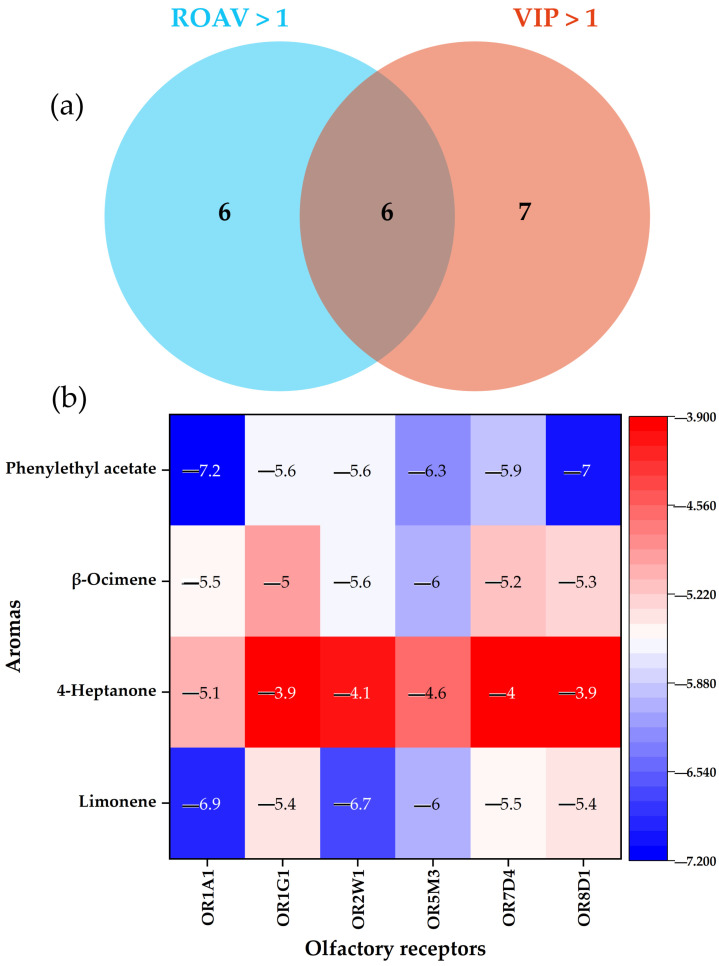
Screening results of key aroma volatile compounds (**a**) and molecular docking results (**b**). ROAV represents relative odor activity value. VIP represents variable importance for the projection value.

**Figure 5 foods-14-00727-f005:**
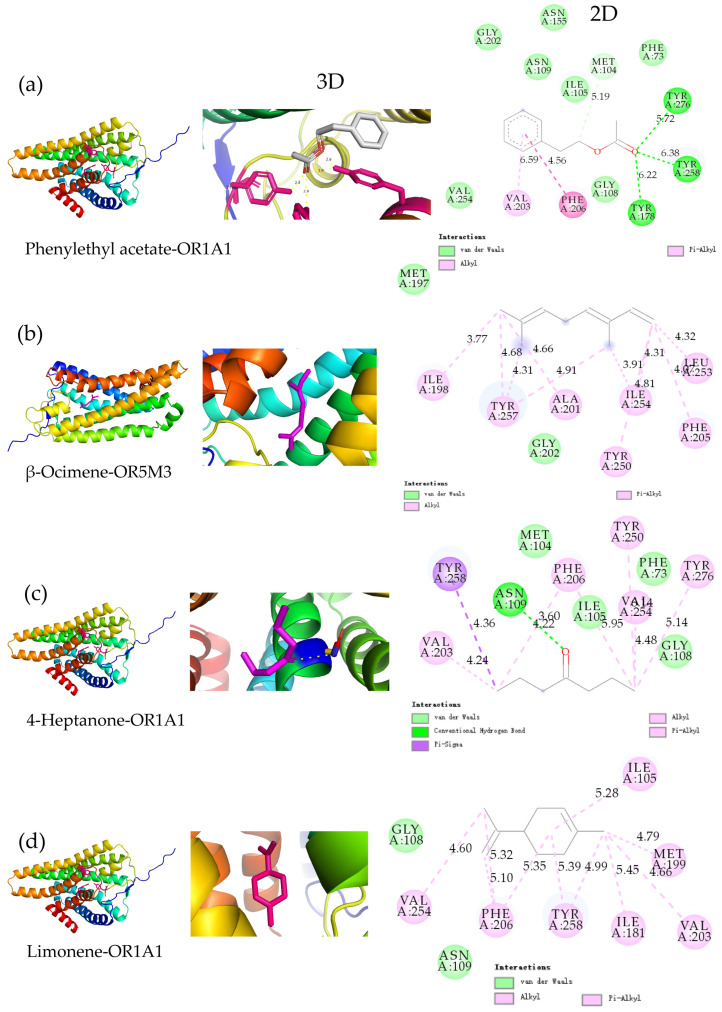
The 3D and 2D plot of MD shows the interaction of 4 key aroma volatile compounds with olfactory receptors. (**a**) Phenylethyl acetate-OR1A1 complex; (**b**) β-Ocimene-OR5M3 complex; (**c**) 4-Heptanone-OR1A1 complex; (**d**) Limonene-OR1A1 complex.

**Table 1 foods-14-00727-t001:** The quantitative results of aroma volatile compounds in pickled radish samples with and without lemongrass essential oil during the 0–28 day storage period.

No.	Aroma Volatile Compounds	CAS	0 Days		14 Days		28 Days	
			PR0	EOPR0	PR14	EOPR14	PR28	EOPR28
Ketones								
1	Propan-2-one	C103457	14.44 ± 0.15 a	13.03 ± 0.46 b	14.39 ± 0.38 a	11.42 ± 0.41 c	15.10 ± 0.63 a	11.25 ± 0.13 c
2	6-Methylhept-5-en-2-one	C110930	0.21 ± 0.02 d	1.19 ± 0.08 c	0.20 ± 0.01 d	1.66 ± 0.13 b	0.24 ± 0.02 d	1.83 ± 0.05 a
3	4-Heptanone	C123193	0.10 ± 0.00 c	0.12 ± 0.01 c	0.12 ± 0.01 c	0.21 ± 0.02 b	0.14 ± 0.02 c	0.40 ± 0.05 a
4	4-Nonanone	C4485090	0.29 ± 0.01 cd	0.27 ± 0.01 d	0.58 ± 0.03 a	0.34 ± 0.03 c	0.26 ± 0.06 d	0.42 ± 0.02 b
5	1-Octen-3-one	C4312996	0.17 ± 0.01 c	0.28 ± 0.01 b	0.32 ± 0.01 b	0.40 ± 0.03 a	0.19 ± 0.02 c	0.43 ± 0.02 a
6	2-Methyl-3-heptanone D	C13019200	12.08 ± 0.21 a	9.88 ± 1.11 bc	11.31 ± 0.45 a	9.15 ± 0.39 cd	10.92 ± 0.82 ab	8.22 ± 0.48 d
7	2-Methyl-3-heptanone M	C13019200	6.66 ± 0.02 a	5.62 ± 0.59 b	5.23 ± 0.17 bc	4.35 ± 0.32 de	4.83 ± 0.21 cd	4.01 ± 0.22 e
8	Cyclopentanone	C120923	0.27 ± 0.03 b	0.23 ± 0.02 cd	0.36 ± 0.02 a	0.25 ± 0.02 bc	0.35 ± 0.01 a	0.21 ± 0.03 d
9	2,3-Butanedione	C431038	0.96 ± 0.03 a	0.94 ± 0.06 a	0.79 ± 0.08 b	0.81 ± 0.05 b	0.83 ± 0.10 b	0.66 ± 0.04 c
Acids								
10	2-Methyl butanoic acid D	C116530	0.39 ± 0.06 d	0.86 ± 0.04 c	0.31 ± 0.02 d	1.41 ± 0.14 a	0.36 ± 0.03 d	1.11 ± 0.05 b
11	2-Methylbutyric acid	C116530	0.76 ± 0.15 a	3.06 ± 2.15 a	2.31 ± 2.41 a	3.04 ± 2.06 a	2.97 ± 2.65 a	2.62 ± 1.29 a
12	3-Methylbutanoic acid	C503742	0.22 ± 0.03 c	0.44 ± 0.01 b	0.21 ± 0.03 c	0.57 ± 0.04 a	0.22 ± 0.03 c	0.47 ± 0.02 b
13	Acetic acid	C64197	25.88 ± 0.38 ab	16.04 ± 0.07 d	24.92 ± 0.79 b	20.75 ± 0.46 c	26.88 ± 1.72 a	21.64 ± 0.16 c
Esters								
14	Butyrolactone	C96480	1.31 ± 0.07 a	1.14 ± 0.03 c	1.22 ± 0.03 b	0.91 ± 0.04 d	1.18 ± 0.04 bc	0.84 ± 0.00 d
15	Lynalyl acetate M	C115957	0.42 ± 0.13 d	2.87 ± 0.10 c	0.44 ± 0.05 d	7.45 ± 0.16 b	0.66 ± 0.08 d	9.47 ± 0.25 a
16	Lynalyl acetate D	C115957	0.14 ± 0.02 c	0.19 ± 0.01 c	0.16 ± 0.04 c	0.58 ± 0.04 b	0.19 ± 0.03 c	0.86 ± 0.05 a
17	Ethyl heptanoate	C106309	0.3 ± 0.03 c	0.46 ± 0.01 b	0.38 ± 0.04 bc	0.45 ± 0.04 b	0.63 ± 0.15 a	0.41 ± 0.02 bc
18	Ethyl crotonate	C623701	0.29 ± 0.02 ab	0.37 ± 0.01 a	0.20 ± 0.04 bc	0.25 ± 0.03 bc	0.39 ± 0.13 a	0.15 ± 0.01 c
19	Phenylethyl acetate M	C111875	0.53 ± 0.07 c	3.34 ± 0.24 b	0.56 ± 0.09 c	3.85 ± 0.17 a	0.71 ± 0.25 c	2.97 ± 0.12 b
20	Phenylethyl acetate D	C13877913	0.31 ± 0.04 c	0.42 ± 0.02 ab	0.34 ± 0.06 c	0.44 ± 0.01 a	0.36 ± 0.03 bc	0.35 ± 0.02 c
21	Ethyl acetate	C141786	10.59 ± 0.13 b	7.01 ± 0.37 de	11.84 ± 0.31 a	6.61 ± 0.24 e	7.47 ± 0.78 cd	8.01 ± 0.16 c
22	ethyl 2-(Methylthio)acetate	C4455134	0.12 ± 0.13 c	0.44 ± 0.03 b	0.05 ± 0.01 c	0.57 ± 0.02 a	0.05 ± 0.01 c	0.62 ± 0.02 a
23	Ethyl propanoate	C105373	0.02 ± 0.00 b	0.03 ± 0.00 b	0.02 ± 0.00 b	0.01 ± 0.00 b	0.07 ± 0.03 a	0.03 ± 0.00 b
Alcohols								
24	2,3-Butanediol	C513859	0.14 ± 0.01 d	1.59 ± 0.06 a	0.17 ± 0.02 d	1.17 ± 0.03 b	0.19 ± 0.06 d	1.05 ± 0.05 c
25	1-Octanol D	C111875	0.05 ± 0.01 d	0.22 ± 0.01 b	0.07 ± 0.01 d	0.27 ± 0.05 a	0.07 ± 0.01 d	0.18 ± 0.01 c
26	3-Methyl-1-butanol	C123513	0.21 ± 0.01 bc	0.19 ± 0.02 c	0.31 ± 0.01 a	0.23 ± 0.02 b	0.24 ± 0.02 b	0.27 ± 0.02 a
27	Octanol D	C13877913	0.13 ± 0.01 c	0.38 ± 0.02 b	0.13 ± 0.00 c	0.57 ± 0.07 a	0.14 ± 0.03 c	0.51 ± 0.04 a
28	2-Methylpropanol	C78831	0.20 ± 0.01 a	0.22 ± 0.01 a	0.24 ± 0.00 a	0.20 ± 0.00 a	0.20 ± 0.00 a	0.21 ± 0.01 a
29	Ethanol	C64175	13.10 ± 0.38 a	13.09 ± 0.37 a	12.10 ± 0.47 b	11.85 ± 0.53 b	13.38 ± 0.49 a	10.74 ± 0.17 c
30	Ethanol M	C64175	0.36 ± 0.01 cd	0.59 ± 0.01 b	0.27 ± 0.02 d	0.48 ± 0.03 bc	0.84 ± 0.20 a	0.39 ± 0.01 cd
Aldehydes								
31	Benzaldehyde	C100527	0.23 ± 0.03 b	2.93 ± 0.12 a	0.22 ± 0.02 b	0.20 ± 0.02 b	0.21 ± 0.02 b	0.16 ± 0.01 b
32	Methional M	C3268493	7.28 ± 0.88 a	4.88 ± 0.13 b	7.25 ± 0.09 a	5.45 ± 0.26 b	6.7 ± 0.23 a	4.95 ± 0.12 b
33	Methional D	C3268493	0.09 ± 0.01 b	0.90 ± 0.03 a	0.10 ± 0.02 b	0.06 ± 0.01 c	0.09 ± 0.01 bc	0.06 ± 0.00 c
34	Nonanal M	C124196	0.57 ± 0.03 d	0.75 ± 0.02 bc	0.84 ± 0.08 ab	0.70 ± 0.06 cd	0.94 ± 0.14 a	0.63 ± 0.03 cd
35	Nonanal D	C124196	0.06 ± 0.01 c	0.07 ± 0.01 b	0.08 ± 0.01 ab	0.07 ± 0.01 bc	0.09 ± 0.01 a	0.06 ± 0.00 c
36	*E*-2-Octenal	C2548870	0.21 ± 0.02 b	3.37 ± 0.34 a	0.24 ± 0.04 b	0.16 ± 0.01 b	0.21 ± 0.02 b	0.16 ± 0.00 b
37	Citronellal	C106230	0.09 ± 0.01 b	1.34 ± 0.17 a	0.12 ± 0.01 b	0.07 ± 0.01 b	0.11 ± 0.02 b	0.07 ± 0.00 b
38	(*E*)-2-Hexenal	C6728263	0.06 ± 0.01b c	0.08± 0.00 b	0.19 ± 0.03 a	0.04 ± 0.00 d	0.20 ± 0.02 a	0.05 ± 0.00 cd
39	Hexanal M	C66251	0.16 ± 0.01 b	0.15 ± 0.01 b	0.56 ± 0.06 a	0.15 ± 0.01 b	0.62 ± 0.07 a	0.12 ± 0.02 b
40	Hexanal D	C66251	0.01 ± 0.00 b	0.01 ± 0.00 b	0.06 ± 0.01 a	0.01 ± 0.00 b	0.07 ± 0.01 a	0.01 ± 0.00 b
41	Heptanal	C111717	0.07 ± 0.01 b	0.08± 0.01 b	0.17 ± 0.01 a	0.09 ± 0.00 b	0.19 ± 0.03 a	0.08 ± 0.00 b
Alkenes								
42	Terpinene	C103457	0.04 ± 0.00 c	0.10 ± 0.02 c	0.05 ± 0.00 c	0.63 ± 0.05 b	0.07 ± 0.01 c	0.70 ± 0.06 a
43	Limonene	C138863	0.07 ± 0.00 d	0.36 ± 0.06 c	0.07 ± 0.01 d	0.60 ± 0.04 a	0.09 ± 0.01 d	0.54 ± 0.03 b
44	β-Ocimene M	C586629	0.06 ± 0.01 c	0.09 ± 0.01 c	0.05 ± 0.00 c	0.66 ± 0.05 b	0.07 ± 0.01 c	0.93 ± 0.05 a
45	β-Ocimene D	C99854	0.05 ± 0.01 c	0.07 ± 0.00 c	0.05 ± 0.00 c	0.26 ± 0.01 b	0.06 ± 0.01 c	0.40 ± 0.03 a
46	Myrcene	C123353	0.02 ± 0.00 b	0.03 ± 0.00 b	0.03 ± 0.00 b	0.08 ± 0.01 a	0.02 ± 0.00 b	0.10 ± 0.01 ab
47	Terpinolene	C67641	0.24 ± 0.01 c	0.23 ± 0.03 c	0.22 ± 0.02 c	0.5 ± 0.04 b	0.14 ± 0.02 d	0.64 ± 0.01 a
Ethers								
48	Diallyl sulfide	C592881	0.05 ± 0.00 c	0.03 ± 0.00 d	0.16 ± 0.01 a	0.03 ± 0.00 d	0.07 ± 0.01 b	0.03 ± 0.00 d

Different lowercase letters in the same row indicate a significant difference (*p* < 0.05). CAS represents chemical abstracts service number. EOPR0, EOPR14, EOPR24, respectively, represent the pickled radish samples of adding lemongrass essential oil for 0, 14, 28 days of storage. PR0, PR14, PR24, respectively, represent the pickled radish samples for 0, 14, 28 days of storage.

**Table 2 foods-14-00727-t002:** Odor thresholds and ROAV of aroma volatile compounds in pickled radish samples with and without lemongrass essential oil during the 0–28 days of the storage period.

No.	Aroma Volatile Compounds	T (µg/kg) [34]	Odorant Description [34]	ROAV					
				PR0	EOPR0	PR14	EOPR14	PR28	EOPR28
1	Acetic acid	1.82	sour, spicy	2.74	1.70	2.64	2.20	2.85	2.29
2	Benzaldehyde	0.35	bitter almond, cherry, nutty	0.13	1.62	0.12	0.11	0.12	0.09
3	Nonanal M	0.0011	rose, citrus, strong oily	100.00	131.58	147.37	122.81	164.91	110.53
4	Nonanal D	0.0011	rose, citrus, strong oily	10.53	12.28	14.04	12.28	15.79	10.53
5	Limonene	0.058	lemon, sweet, orange, pine oil	0.23	1.20	0.23	2.00	0.30	1.80
6	4-Heptanone	0.0082	Fruity	2.35	2.82	2.82	4.94	3.29	9.41
7	Phenylethyl acetate M	0.25	citrus, sweet, rose, honey, fruity	0.41	2.58	0.43	2.98	0.55	2.30
8	Citronellal	0.0052	lemon, lemongrass, rose	3.34	49.73	4.45	2.60	4.08	2.60
9	4-Nonanone	0.0082	green	6.82	6.35	13.65	8.00	6.12	9.88
10	β-Ocimene M	0.034	fresh woody, sweet, citrus	0.34	0.51	0.28	3.75	0.40	5.28
11	β-Ocimene D	0.034	fresh woody, sweet, citrus	0.28	0.40	0.28	1.48	0.34	2.27
12	2,3-Butanedione	0.059	butter, popcorn, sweet taste, sour rice	3.14	3.07	2.58	2.65	2.71	2.16

T represents odor thresholds. ROAV represents relative odor activity value. EOPR0, EOPR14, EOPR28, respectively, represent the pickled radish samples of adding lemongrass essential oil for 0, 14, 28 days of storage. PR0, PR14, PR28, respectively, represent the pickled radish samples for 0, 14, 28 days of storage.

## Data Availability

The original contributions presented in this study are included in the article/Appendix A. Further inquiries can be directed to the corresponding authors.

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
