# Peer review of "Effects of Lemongrass Essential Oil on Key Aromas of Pickled Radish During Storage Using HS–GC–IMS and in Silico Approaches"

_foods, 2025, doi:10.3390/foods14050727_

Round 1
Reviewer 1 Report
Comments and Suggestions for Authors
The manuscript "Effects of Lemongrass Essential Oil on Key Aromas of Pickled Radish During Storage Using HS-GC-IMS and In Silico Approaches" is interesting and presents valuable findings on the effects of lemongrass essential oil on pickled radish aroma during storage. The introduction is well written, and the results are appropriately discussed. However, some sections, particularly in Materials and Methods, as well as figure and table descriptions, require improvement for clarity and completeness. Below are detailed suggestions for enhancing the manuscript:
Materials and Methods: Please, provide a more detailed description of the experimental design to ensure reproducibility:
- Clarify the number of replicates for each treatment group. Specify the quantities of pickled radish and lemongrass essential oil used per treatment replicates.
- Describe the storage conditions comprehensively: state the temperature, storage duration, and type of containers used.
Statistical Analysis (Line 149): Explicitly clarify whether the experiments involved exactly three replicates or a minimum of three replicates to avoid ambiguity.
Volatile compounds terminology: replace the term "aromas" with "aroma volatile compounds" throughout the manuscript to maintain scientific precision (e.g., Line 114 but it should be done in whole manuscript). Moreover, italicize E and Z isomers when referencing volatile compounds and ensure lowercase formatting for volatile compound names (e.g., Line 284).
Figures and Tables:
- Make figure legends more descriptive and self-explanatory. Include details about the treatments, sample groups, and any abbreviations used in the figures and tables.
- Figures: Revise figure titles to provide greater clarity and improve the description for Figures to explain its relevance.
- Ensure also that all tables have clear and complete titles and legends. Include the following details: number of samples analyzed. statistical comparison methods, full explanations for any abbreviations or acronyms used in the table.
In my opinion, the manuscript contributes to the field of food science, particularly for researchers investigating natural preservatives and aroma volatile compounds. Implementing these suggestions will improve the clarity and precision of the manuscript, ensuring readers can easily follow the experimental design and results.
Author Response
Dear Reviewer,
We sincerely appreciate your efforts for our manuscript "Effects of Lemongrass Essential Oil on Key Aromas of Pickled Radish During Storage Using HS-GC-IMS and In Silico Approaches" (ID: foods-3371039) and the opportunity you have given us to further improve it.
We appreciate Reviewer’s very kind comments. We have revised the paper according to those comments and respond to the comments one by one and carefully proof-read the manuscript to minimize misprints errors. We are convinced that the comments of reviewer improved the paper.
The major changes are as follows:
The manuscript "Effects of Lemongrass Essential Oil on Key Aromas of Pickled Radish During Storage Using HS-GC-IMS and In Silico Approaches" is interesting and presents valuable findings on the effects of lemongrass essential oil on pickled radish aroma during storage. The introduction is well written, and the results are appropriately discussed. However, some sections, particularly in Materials and Methods, as well as figure and table descriptions, require improvement for clarity and completeness. Below are detailed suggestions for enhancing the manuscript:
Materials and Methods: Please, provide a more detailed description of the experimental design to ensure reproducibility:
-Clarify the number of replicates for each treatment group. Specify the quantities of pickled radish and lemongrass essential oil used per treatment replicates.
Thanks to the advice, there are three replicates for each group. The quantity of pickled radish in each group is 300 g, and the concentration of lemongrass essential oil is 0.1% (w/w). Which are written in red color font in the text.
L 80-82:
“Each group's 1 kg sample was randomly and equally apportioned into three replicate tests. Among them, three group samples were treated with 0.1% (w/w) of lemongrass essential oil for 28 days storage experiment.”
-Describe the storage conditions comprehensively: state the temperature, storage duration, and type of containers used.
Thanks to the advice, the storage conditions have been provided. Which are written in red color font in the text.
L 82-89:
“During the 0-28 day storage period, the samples were packaged in food-grade vacuum bags and stored at room temperature. Samples, both with and without added lemongrass essential oil, were collected every 14 days and subsequently rapidly frozen in liquid nitrogen for further analysis. And, the samples containing lemongrass essential oil were respectively labeled as EOPR0, EOPR14, and EOPR28, while the other samples were respectively marked as PR0, PR14, and PR28.”
Statistical Analysis (Line 149): Explicitly clarify whether the experiments involved exactly three replicates or a minimum of three replicates to avoid ambiguity.
Thanks to the advice, the experiments involved exactly three replicates, we have been revised it. Which are written in red color font in the text.
L 153:
“All experiments were performed with three independent biological replicates.”
Volatile compounds terminology: replace the term "aromas" with "aroma volatile compounds" throughout the manuscript to maintain scientific precision (e.g., Line 114 but it should be done in whole manuscript). Moreover, italicize E and Z isomers when referencing volatile compounds and ensure lowercase formatting for volatile compound names (e.g., Line 284).
Thanks to the advice, the term "aromas" has been replaced to the "aroma volatile compounds" throughout the manuscript. And, E and Z isomers have been revised to the italics, the volatile compound names in line 284 have been revised to lowercase formatting. Which are written in red color font in the text.
L 117
“2.3.3. Qualitative and quantitative of aroma volatile compounds.”
L 299-301
“The loading plot of PCA indicated that compounds butyrolactone, 2,3-butanedione, 2,3-butanediol, and acetic acid might be accountable for this difference (Figure 3b).”
Figures and Tables:
-Make figure legends more descriptive and self-explanatory. Include details about the treatments, sample groups, and any abbreviations used in the figures and tables.
Thanks to the advice, all the figures and tables have been detailed to make them more descriptive and self-explanatory. Which are written in red color font in the text.
-Figures: Revise figure titles to provide greater clarity and improve the description for Figures to explain its relevance.
Thanks to the advice, the figure titles have been revised. Which are written in red color font in the text.
L 197-202:
“Figure 1. Analysis of aroma volatile compounds in different pickled radish samples. (a) Two-dimensional spectrum of GC-IMS; (b) topographic subtraction map; (c) three-dimensional topographic map. EOPR0, EOPR14, EOPR24 respectively represents the pickled radish samples of adding lemongrass essential oil for 0, 14, 28 days storage. PR0, PR14, PR24 respectively represents the pickled radish samples for 0, 14, 28 days storage.”
L 288-292:
“Figure 2. The results of fingerprint spectra analysis of with and without lemongrass essential oil pickled radish samples during the 0-28 days storage period. EOPR0, EOPR14, EOPR24 respectively represents the pickled radish samples of adding lemongrass essential oil for 0, 14, 28 days storage. PR0, PR14, PR24 respectively represents the pickled radish samples for 0, 14, 28 days storage.”
L 317-322:
“Figure 3. The results of multivariate statistical analysis. (a) PCA score plot; (b) loading plot; (c) OPLS-DA score plot; (d) permutation test of OPLS-DA model; (e) variable importance for the projection value (VIP) values. EOPR0, EOPR14, EOPR24 respectively represents the pickled radish samples of adding lemongrass essential oil for 0, 14, 28 days storage. PR0, PR14, PR24 respectively represents the pickled radish samples for 0, 14, 28 days storage.”
L 378-380:
“Figure 4. Screening results of key aroma volatile compounds (a) and molecular docking results (b). ROAV represents relative odor activity value. VIP represents variable importance for the projection value.”
-Ensure also that all tables have clear and complete titles and legends. Include the following details: number of samples analyzed. statistical comparison methods, full explanations for any abbreviations or acronyms used in the table.
Thanks to the advice, all titles and legends of tables have been improved, including the number of samples analyzed, statistical comparison methods, full explanations for any abbreviations and acronyms used in the table.
L 257-258:
“Table 1. The quantitative results of aroma volatile compounds in with and without lemongrass essential oil pickled radish samples during the 0-28 days storage period.”
L 259-262:
“Different lowercase letters in the same row indicate a significant difference (p < 0.05). CAS represents chemical abstracts service number. EOPR0, EOPR14, EOPR24 respectively represents the pickled radish samples of adding lemongrass essential oil for 0, 14, 28 days storage. PR0, PR14, PR24 respectively represents the pickled radish samples for 0, 14, 28 days storage.”
L 354-355:
“Table 2. Odor thresholds and ROAV of aroma volatile compounds in with and without lemongrass essential oil pickled radish samples during the 0-28 days storage period.”
L 356-358:
“T represents odor thresholds. ROAV represents relative odor activity value. EOPR0, EOPR14, EOPR24 respectively represents the pickled radish samples of adding lemongrass essential oil for 0, 14, 28 days storage. PR0, PR14, PR24 respectively represents the pickled radish samples for 0, 14, 28 days storage.”
Thank you again for the kind advices.
Best regards,
Dr. Zelin Li
Agro-Products Processing Research Institute, Yunnan Academy of Agricultural Sciences, Kunming 650223, China.
E-mail: lzl1054994094@126.com

Reviewer 2 Report
Comments and Suggestions for Authors
The manuscript studies the effect of lemongrass essential oil on the key aroma compounds of pickled radish during storage, using diverse laboratory and statistical techniques. To emphasise using molecular docking to investigate the molecular mechanism through which lemongrass essential oil enhances the fruity and sweet aroma during storage. The manuscript is interesting but requires some refinements: the experimental design needs clarification and requires homogenisation of some terms and capital letters utilisation. Also, some clarification regarding the selection of relevant compounds would be acknowledged. Some specific comments follow.
Specific
L15 lemongrass. Please homogenise
L20 p < 0.05
L21 projection < 1 , six potential …
L64 lemongrass
L80-81. Please homogenise the use of lemongrass throughout the manuscript.
L78-79. The sentence is vague. Please specify if you used one sample of 6 kg or several of the same size or different sizes. The sentence requires clarification.
L80. Three parts? Three groups? Three samples?
L81. Were samples treated three times?
L78-84. The whole sentence needs clarification.
L89. Usually, a text space is situated before and after ≥ or similar symbols.
L98. The sentence requires clarification.
L104. The verb is missing
Line 111. , respectively. Please revise the syntax and grammar of the entire manuscript.
L119. Eq. 1
L127. Please use the full name for this and all the abbreviations used in the manuscript when mentioned first.
L137. Please, full name for MD
L256-260. The sentence requires clarification. What is the meaning of each cell?
L161. Do you mean spoilage?
Fig 1. The full names of all abbreviations should be explained in the legend. Most readers are not familiar with them. Also, (Figure 1c) should be relocated.
Fig 1. This is only a figure. Then, groups a, b, and c should be defined in other terms and identified conveniently in the figure legend.
Table 1 Supplementary. Initial capital letters (or not) for compounds should be homogenised.
L284. Revise capital letter in compound names
L293 “ a total “ could be removed
L293-296. According to Figure 3, this affirmation is questionable since there is no VIP significantly higher than 1, which is consistently within the confidence limits of this parameter. Please explain
L 306-307. How do you consider VIP > 1? Simply higher than 1 or significantly higher than 1?
Author Response
Dear Reviewer,
We sincerely appreciate your efforts for our manuscript "Effects of Lemongrass Essential Oil on Key Aromas of Pickled Radish During Storage Using HS-GC-IMS and In Silico Approaches" (ID: foods-3371039) and the opportunity you have given us to further improve it.
We appreciate Reviewer’s very kind comments. We have revised the paper according to those comments and respond to the comments one by one and carefully proof-read the manuscript to minimize misprints errors. We are convinced that the comments of reviewer improved the paper.
The major changes are as follows:
The manuscript studies the effect of lemongrass essential oil on the key aroma compounds of pickled radish during storage, using diverse laboratory and statistical techniques. To emphasise using molecular docking to investigate the molecular mechanism through which lemongrass essential oil enhances the fruity and sweet aroma during storage. The manuscript is interesting but requires some refinements: the experimental design needs clarification and requires homogenisation of some terms and capital letters utilisation. Also, some clarification regarding the selection of relevant compounds would be acknowledged. Some specific comments follow.
Specific
L15 lemongrass. Please homogenise.
Thanks to the advice, the lemongrass has been uniformed in the whole text. Which are written in red color font in the text.
L 15-16:
“To investigate the effects of lemongrass essential oil on the key aroma volatile compounds of pickled radish (PR) during storage”
L20 p < 0.05.
Thanks to the advice, the “p < 0.05” has been uniformed in the whole text. Which are written in red color font in the text.
L 20-21:
“Using the screening criteria of p < 0.05 and variable importance for the projection > 1 in multivariate statistical analysis”
L21 projection < 1 , six potential ….
Thanks to the advice, the “projection < 1 , six potential” have been revised. Which are written in red color font in the text.
L 22:
“activity value > 1, six potential key aroma compounds were selected. ”
L64 lemongrass
Thanks to the advice, the lemongrass has been uniformed in the whole text. Which are written in red color font in the text.
L 64-66:
“The volatile components of lemongrass essential oil not only possess the aforesaid biological activities but also endow it with its aromatic features.”
L80-81. Please homogenise the use of lemongrass throughout the manuscript.
Thanks to the advice, the lemongrass has been uniformed in the whole text. Which are written in red color font in the text.
L 82:
“with 0.1% (w/w) of lemongrass essential oil for 28 days storage experiment.”
L78-79. The sentence is vague. Please specify if you used one sample of 6 kg or several of the same size or different sizes. The sentence requires clarification.
Thanks to the advice, the used samples are same size and the sentence has been rewritten. Which are written in red color font in the text.
L 78-79:
“A total of 6 kg of pickled radish samples with same size were provided by Honghe Hongbin Food Co., Ltd. on June 12, 2024.”
L80. Three parts? Three groups? Three samples?
Thanks to the advice, this is three groups. Which are written in red color font in the text.
L 81-82:
“Among them, three group samples were treated with 0.1% (w/w) of lemongrass essential oil for 28 days storage experiment.”
L81. Were samples treated three times?
Thanks to the advice, the samples were uniformly treated and then randomly grouped.
L78-84. The whole sentence needs clarification.
Thanks to the advice, the whole sentence has been revised. Which are written in red color font in the text.
L 78-88:
“A total of 6 kg of pickled radish samples with same size were provided by Honghe Hongbin Food Co., Ltd. on June 12, 2024. These samples were randomly divided into 6 groups, each weighing 1 kg. Each group's 1 kg sample was randomly and equally apportioned into three replicate tests. Among them, three group samples were treated with 0.1% (w/w) of lemongrass essential oil for 28 days storage experiment. During the 0-28 day storage period, the samples were packaged in food-grade vacuum bags and stored at room temperature. Samples, both with and without added lemongrass essential oil, were collected every 14 days and subsequently rapidly frozen in liquid nitrogen for further analysis. And, the samples containing lemongrass essential oil were respectively labeled as EOPR0, EOPR14, and EOPR28, while the other samples were respectively marked as PR0, PR14, and PR28.”
L89. Usually, a text space is situated before and after ≥ or similar symbols.
Thanks to the advice, a text space has been added in the whole text. Which are written in red color font in the text.
L 93-94:
“Nitrogen (purity ≥ 99.999%) and 20 mL of headspace bottle were obtained from Shandong Hanon Scientific Instruments Co.,Ltd (Shandong, China).”
L98. The sentence requires clarification.
Thanks to the advice, this sentence has been deleted.
L104. The verb is missing
Thanks to the advice, the verb has been added. Which are written in red color font in the text.
L 107-109:
“The GC conditions were as follows: the injection volume was 500 µL. Nitrogen (with a purity of no less than 99.999%) was employed as the carrier gas.”
Line 111. , respectively. Please revise the syntax and grammar of the entire manuscript.
Thanks to the advice, the syntax and grammar of the entire manuscript have been revised. Which are written in red color font in the text.
L 114-115:
“tube were 53 mm and 45°C, respectively. The intensity of the electric field was set at 500 V/cm in the positive mode.”
L119. Eq. 1
Thanks to the advice, this sentence has been revised. Which are written in red color font in the text.
L 121:
“The RI was calculated as presented in Eq. 1.”
L127. Please use the full name for this and all the abbreviations used in the manuscript when mentioned first.
Thanks to the advice, the full name of the abbreviation in L 127 has been provided in the Introduction section.
L 69-73:
“Therefore, this study is aimed at comparing and screening the effects of lemongrass essential oil on the key aroma and its flavor mechanism of PR during storage through the utilization of HS-GC-IMS, fingerprint analysis, multivariate statistical analysis, relative odor activity value (ROAV), and molecular docking (MD) methods.”
L137. Please, full name for MD
Thanks to the advice, the full name of the abbreviation in L 137 has been provided in the Introduction section.
L 69-73:
“Therefore, this study is aimed at comparing and screening the effects of lemongrass essential oil on the key aroma and its flavor mechanism of PR during storage through the utilization of HS-GC-IMS, fingerprint analysis, multivariate statistical analysis, relative odor activity value (ROAV), and molecular docking (MD) methods.”
L256-260. The sentence requires clarification. What is the meaning of each cell?
Thanks to the advice, we did not locate the phrase "each cell" in the sentences from lines 256 to 260. Should there be any additional comments of reviewer, we are fully prepared to undertake further revisions to enhance the quality of our manuscript.
L161. Do you mean spoilage?
Thanks to the advice, the L 161 means that “To explore the alterations of key aromas in diverse samples during storage, GC-IMS was employed to detect volatile compounds at various stages.”
Fig 1. The full names of all abbreviations should be explained in the legend. Most readers are not familiar with them. Also, (Figure 1c) should be relocated.
Thanks to the advice, the full names of all abbreviations have been added and Figure 1c has been revised. Which are written in red color font in the text.
L 197-202:
“Figure 1. Analysis of aroma volatile compounds in different pickled radish samples. (a) Two-dimensional spectrum of GC-IMS; (b) topographic subtraction map; (c) three-dimensional topographic map. EOPR0, EOPR14, EOPR24 respectively represents the pickled radish samples of adding lemongrass essential oil for 0, 14, 28 days storage. PR0, PR14, PR24 respectively represents the pickled radish samples for 0, 14, 28 days storage.”
Fig 1. This is only a figure. Then, groups a, b, and c should be defined in other terms and identified conveniently in the figure legend.
Thanks to the advice, this figure has been revised.
Table 1 Supplementary. Initial capital letters (or not) for compounds should be homogenised.
Thanks to the advice, all initial letters have been revised to uppercase letters. Which are written in red color font in the text.
L284. Revise capital letter in compound names
Thanks to the advice, the capital letter in compound names have been revised. Which are written in red color font in the text.
L 300-301:
“butyrolactone, 2,3-butanedione, 2,3-butanediol, and acetic acid might be accountable for this difference (Figure 3b).”
L293 “ a total “ could be removed
Thanks to the advice, “ a total “ has been removed. Which are written in red color font in the text.
L 309-310:
“As shown in Figure 3e, 13 volatile aromatic compounds were screened out, including ethyl propanoate, ethanol M,”
L293-296. According to Figure 3, this affirmation is questionable since there is no VIP significantly higher than 1, which is consistently within the confidence limits of this parameter. Please explain
Thanks to the advice, it merely means that the VIP is simply greater than 1, rather than significantly greater than 1.
L 306-307. How do you consider VIP > 1? Simply higher than 1 or significantly higher than 1?
Thanks to the advice, it merely means that the VIP is simply greater than 1, rather than significantly greater than 1.
Thank you again for the kind advices.
Best regards,
Dr. Zelin Li
Agro-Products Processing Research Institute, Yunnan Academy of Agricultural Sciences, Kunming 650223, China.
E-mail: lzl1054994094@126.com
